# Systems Medicine Approach for Tinnitus with Comorbid Disorders

**DOI:** 10.3390/nu14204320

**Published:** 2022-10-15

**Authors:** Birgit Mazurek, Matthias Rose, Holger Schulze, Christian Dobel

**Affiliations:** 1Tinnitus Center, Charité-Universitätsmedizin Berlin, 10117 Berlin, Germany; 2Medical Department, Division of Psychosomatic Medicine, Charité-Universitätsmedizin Berlin, 10117 Berlin, Germany; 3Department of Otorhinolaryngology–Head and Neck Surgery, Universitätsklinikum Erlangen, 91054 Erlangen, Germany; 4Department of Otorhinolaryngology, Jena University Hospital, 07743 Jena, Germany

**Keywords:** tinnitus, depression, pain, comorbidity, system medicine

## Abstract

Despite the fact that chronic diseases usually occur together with a spectrum of possible comorbidities that may differ strongly between patients, they are classically still viewed as distinct disease entities and, consequently, are often treated with uniform therapies. Unfortunately, such an approach does not take into account that different combinations of symptoms and comorbidities may result from different pathological (e.g., environmental, genetic, dietary, etc.) factors, which require specific and individualised therapeutic strategies. In this opinion paper, we aim to put forward a more differentiated, systems medicine approach to disease and patient treatment. To elaborate on this concept, we focus on the interplay of tinnitus, depression, and chronic pain. In our view, these conditions can be characterised by a variety of phenotypes composed of variable sets of symptoms and biomarkers, rather than distinct disease entities. The knowledge of the interplay of such symptoms and biomarkers will provide the key to a deeper, mechanistic understanding of disease pathologies. This paves the way for prediction and prevention of disease pathways, including more personalised and effective treatment strategies.

## 1. Introduction

Traditional disease models and accepted manuals, such as the International Classification of Diseases (ICD) and the Diagnostic and Statistical Manual of Mental Disorders (DSM), currently describe chronic diseases as distinct disease entities. However, most chronic comorbid conditions present multifactorial causes and share similar determining factors, often originating from lifestyle characteristics (e.g., dietary behaviour or physical activity). Acknowledging the multifactorial and interactive nature of biological, social, and psychological factors, particularly in psychiatric conditions, it was proposed to use the term disorder rather than disease [1]. Additionally, disorder is considered less stigmatising than disease. Using this background, tinnitus is characterised by an auditory phenomenon, i.e., the phantom sound, in combination with accompanying affective components [1]. The perception of a distressing sound like tinnitus could be due to changes in the cortical structure of a patient’s brain, possibly provoked by comorbid hearing loss or by depression-associated pathological brain activity. With regard to other diseases and disorders like cancer, respiratory diseases, or diabetes, it is well established that the most prevalent chronic comorbid conditions are characterised by a multifactorial aetiology, with no single factor sufficiently explaining their symptomatic presentation and their future development [2]. Therefore, the current practice of diagnosing disorders in isolation poses a severe hurdle for effective treatment. This is particularly problematic in the acute stages of a pathology [3,4]. Nevertheless, clinicians still largely operate within such disease models, leading to isolated clinical guidelines for chronic comorbid diseases, failing to take their interconnectivity sufficiently into account. To close this gap in the understanding of chronic comorbid conditions, a substantially different disease model is required. By viewing comorbid “disorders” as “collections of symptoms” of a disturbed system, we may be better able to understand their often-shared origins and, most importantly, treat them optimally. For example, if a patient presents with fever and bloody cough, those two evident symptoms would not be classified as “comorbid” conditions of tuberculosis, but the result of a shared underlying pathology of the disease. Therefore, we advocate here for a paradigm shift—from applying isolated, disorder-specific categories, thereby neglecting individual characteristics, to using system-based, empirically defined, and connected phenotypes to facilitate prediction, prevention, personalisation and participation of the individual patient (also termed P4 medicine; [5]). This constitutes a shift from reactive to prospective medicine and from disease to health. Modern technologies like genomics or artificial intelligence (AI) are able to support this shift.

For many years, high-throughput technologies, e.g., in genomics, produced massive amounts of data, which have led to limited clinical breakthroughs as of yet. Positive exceptions include rare diseases, pharmacogenetics and, of course, oncology. With the advent of modern algorithms and resulting technologies of AI, clinical breakthroughs in other domains are expected in the near future. The advanced monitoring of individuals with precision medicine, e.g., in terms of their lifestyle and resulting parameters, has now become an option, which enables us to identify early warning signs that allow for an earlier personalised beginning of interventions on the part of the treating clinicians. In addition, by creating conscious awareness of their individual risk factors and lifestyle choices (e.g., nutrition, tobacco, physical exercise, socio-economic status, early life events), this presents an opportunity to empower patients towards lasting behavioural changes to prevent the onset, further progression, and/or chronic manifestation of potential disorders [2].

Here, we will elaborate this paradigm shift by focussing on the example of three functional conditions: tinnitus, depression, and chronic pain. Instead of focussing on diseases and disorders with obvious structural aberrations and damage, this particular choice will enable understanding of those pathologies in the light of the dynamic nature of a systems medicine approach. Recent research findings support the theory of a common physiological and pathophysiological basis for these three widespread disorders with both mental and physical influences. We will therefore discuss all three conditions as “collections of symptoms” of a shared underlying pathological development. Thus, all three indicate a disturbed system, which may also serve as an early indicator for the development of other chronic conditions in the long-term. This will allow an investigation of shared common causal mechanisms and discriminative patterns using big data analytics and AI technologies. Due to the inherent complexity of this problem, it is helpful to start with a small set of reference conditions and use cross-validation to generalise strategies thereafter, in order to expand them to other medical areas. Tinnitus, depression, and chronic pain are ideal candidates to demonstrate this paradigm-shift. All three are highly prevalent in the general population (see Table 1) as well as commonly comorbid (see Figure 1). As an example, in a European multicentre cohort of tinnitus patients, 43% of respondents reported their tinnitus as a big or very big problem, showing a strong relation to depressive symptoms and, connectedly, a lower quality of life [6]. As outlined in Table 1, these impairments represent an enormous burden to patients, their social circle, and society in general.

Taken together, tinnitus, depression, and chronic pain affect at least a quarter of the European population, and have a cumulated societal cost between EUR 400 billion and over EUR 700 billion annually in the European Union. These costs can be significantly reduced by prediction and prevention, as proposed by our approach [7,8,9,10,11,12,13].

Therefore, we will here use those three disorders as prototype conditions to identify causal mechanisms that determine disorder progression towards combinations of chronic and associated pathologies. This approach only became feasible now, since new scientific developments allow for the measurement of biological processes on a molecular and cellular level in unprecedented detail, e.g., using high-throughput mass spectrometry. In addition, the ubiquitous use of smartphones simplifies behavioural tracking as mobile app technologies are being utilised for research. Finally, advances in big data analysis and AI based machine learning allow for the integration of such rich and heterogeneous datasets in order to model both phenotype characteristics and predictors for health trajectories. These new possibilities will advocate a paradigm shift towards the identification and treatment of phenotypes that are characterised by varying biopsychological expressions of disturbed underlying systems. Thereafter, the established framework can be generalised towards other disorders.

## 2. Comorbidity in Chronic Tinnitus and Its Challenge to the Health Care Systems

Despite their classical separation into multiple categories, tinnitus, depression, and chronic pain share common and specific phenomenological [14] and neurophysiological variance [15], suggesting an underlying physiological and pathophysiological basis (Figure 1). As detailed below, analysis of existing databases for comorbid disorders allows identifying (a) phenotypical overlap, (b) common risk profiles, and (c) gaps in current data collection.

### 2.1. A Common Basis of Tinnitus and Depression

On a phenomenological level, emotional and cognitive difficulties can precede, exacerbate, or result from the tinnitus percept [16]; studies point to depressive mood as a key factor in rendering tinnitus distressing [17]. Studies have highlighted the effect of acute stress on auditory processing [18] via processes that have also been implicated in anxiety or depression, such as changes in attention and cognition [19,20], changes in cortisol levels [21], or limbic processes [22]. This suggests distress as a shared risk factor for the development, exacerbation, or maintenance of both tinnitus and emotional difficulties, particularly depression. Focussing on shared factors between tinnitus and depression, another study identified distress as a mediating variable that accounted for the relationship between tinnitus and concurrent mood disorder [23]. Along these lines, several studies have highlighted a key role of depression in the development of tinnitus. For example, a recent study demonstrated that the relationship between tinnitus handicap and anxiety was fully mediated by depressive symptomatology—thus highlighting shared variance between these constructs whilst suggesting depressive symptoms as a key factor in predicting those [24]. Further highlighting the role of depression, a recent prospective study found that hearing loss and depression predicted subsequent subjective tinnitus loudness [25], whilst a reduction of depressive mood predicted subsequent improvements in a representative Swedish sample [26].

### 2.2. A Common Basis of Tinnitus and Chronic Pain

Tinnitus and chronic pain share many commonalities regarding development, causes, phenomenology, and treatment. Despite the considerable co-occurrence of tinnitus and chronic pain [27,28,29,30,31,32,33], a comprehensive approach in research and management is rarely taken. Both chronic pain and tinnitus are subjective, multifactorial-influenced sensations [34,35], and are often associated with hypersensitivity to sensory stimuli initially thought to originate from peripheral receptors: auditory for tinnitus (hyperacusis) and tactile for chronic pain (allodynia and hyperalgesia) [34]. The recent identification of chronic pain receptors in the hearing organ, the cochlea, stresses the commonalities [36]. However, a central origin playing a causal role in the development and maintenance of both dysfunctions is the more likely implication [31,37,38].

Beyond potential sensory or (neuro-)physiological contributors [37,39], cognitive-affective processes are known to play key roles in the subjective experience, maintenance, and potential chronic manifestation of each syndrome [40,41]. As an example, patients with chronic tinnitus experience considerable emotional distress [42,43], and report high levels of depression [44,45], anxiety [46], and other somatoform symptoms [47,48], constituting the phenomena of decompensated (vs. compensated) tinnitus. Analogously, chronic pain is frequently accompanied by psychopathological emotional symptoms [49,50,51], and a substantial body of work has highlighted interactions of cognitive and affective factors in mediating experiences of pain sensations [40,52,53,54,55,56,57,58,59]. Consequently, regarding treatment, tinnitus or chronic pain related affective symptoms are key targets of psychological or multimodal interventions that have been shown to be effective [47,60,61].

### 2.3. A Common Basis of Chronic Pain and Depression

Closing the circle, chronic pain is commonly accompanied by depression [62]. The prevalence of chronic pain in depressed cohorts as well as depression in chronic pain cohorts are higher than in the normal population [62]. Moreover, when chronic pain is moderate to severe, impairs function, or is refractory to treatment, it is associated with more depressive symptoms and worse outcomes, such as decreased work function, increased health care utilisation, or lower quality of life [62]. Depression is one of the strongest predictors of low back pain, and the intensity of back pain has been shown to correlate with the severity of depression [63]. In a multicentre cohort study, major depressive episodes were amongst the most prevalent comorbidities [64]. Multiple studies suggest an overlapping neural network for processing of pain and emotion [65,66], and functional and morphological changes as one basis for both pathologies [67]. Studies have also found overlaps between pain- and depression-induced neuroplastic changes [68]. Hence, the processing of nociceptive input is altered in depressed patients.

Taken together, as exemplified by the strong overlap between tinnitus, depression and chronic pain, tinnitus should not be considered as a homogeneous, unitary disorder entity. This is supported by findings from neurophysiological studies.

## 3. Tinnitus and Comorbid Conditions as Neural Network Disorders

Advances in neuroscientific, behavioural, and clinical research have shown that tinnitus, depression, anxiety, and chronic pain are neural network disorders in which sensory, limbic/emotional, and cortical/attention-related networks are implicated [26,29,69]. The nuanced connections between the frontal cortex and limbic structures are well established, in addition to their role in clinical or subclinical affective states [35,36]. Functional connectivity, as a powerful method to assess the interplay of brain regions, describes the temporal dependencies of neuronal activity patterns of anatomically separated regions [38] in response to a stimulus or task, as well as in a resting state. It tries to assess how well different regions are connected, and how well information from one region is transferred to another. Functional connectivity studies implicated that compared to controls, tinnitus patients showed decreased connectivity between the auditory cortices and fronto-parietal, anterior cingulate, and subcortical regions [37,39,40]. Similarly, a review on resting state fMRI connectivity underlying tinnitus further indicated that areas such as the posterior cingulate cortex, the medial prefrontal cortex, inferior frontal cortex, medial prefrontal cortex, insula, and para-hippocampus appear differentially connected in tinnitus patients compared to controls [34]. Moreover, one study showed increased connectivity between the auditory cortex and the amygdala in patients with tinnitus [41]. However, the exact nature of the interplay of frontal and limbic systems in the generation or maintenance of chronic tinnitus is still under debate [70]. Additionally, research suggests that an interaction of maladaptive plasticity and dysfunctional learning processes could be partially responsible for the generation and chronic manifestation of tinnitus [42,43]. This model suggests that hearing loss and the resulting deafferentation of corresponding cortical structures may generate the “sound” of tinnitus. Subsequently, learned attentional and emotional processes may lead to chronic tinnitus.

## 4. Comorbidity as a Collection of Symptoms

In this paper, we consider tinnitus, depression, and chronic pain as network disorders, i.e., there is a high correlation and dependency of symptoms. Consequently, to diagnose and treat them with regard to only one disorder, as is currently done in most instances, is a futile approach. As briefly reviewed above, our approach is supported by recent research from neuroscientific studies demonstrating that tinnitus, depression, and chronic pain all involve sensory, limbic/emotional, and cortical/attention-related networks [38,65,66,71,72]. This applies for both clinical and subclinical conditions [73,74,75]. The involvement of various physiological, sensory, emotional, and cognitive domains is supported by clinical findings, stressing the high co-occurrence of dysfunctions in several domains [19,44].

Disorders with high comorbidity pose special problems to health care systems, from prevention to diagnosis and treatment. This is due to various reasons. Disorders with high comorbidity are characterised by a multitude of symptoms. Some of them are only subjectively perceivable (amount of pain, tinnitus), while others can be measured objectively. Therefore, identification of these disorders depends strongly on health literacy and communicative abilities from patients and physicians. Often patients and physicians weigh symptoms differently, and as a consequence, important aspects of disorders and collections of symptoms are neglected or not communicated at all [76].

In this context, a gap in the current knowledge should be kept in mind: while recent research, as illustrated above, points towards shared pathophysiological and pathopsychological processes underlying all three disorders, the implications for prevention, prediction, personalised diagnosis and treatment, as well as participation of the patient are not specified [68]. To close this gap, the analysis of combined, large databases with modern technological approaches to pinpoint a common risk profile and to predict and prevent chronic manifestation is required.

## 5. Identification of Phenotypes

Thus far, comprehensive data, if available, have only been analysed separately from each other. For phenotyping in a strictly data driven approach, we propose the following steps. In order (1) to reach a sufficiently large population for new types of analyses and (2) to be able to estimate the impact of lifestyle, social, and environmental factors (such as nutrients, nutrition, alcohol, socio-economic status, tobacco, physical activity) that may influence the development of pathological processes, the information from such datasets should be combined across centres. This work requires the collaboration of clinical and research partners in a national or even multinational approach.

We propose that the first step is the establishment of novel phenotypes (Figure 2) [a] within each disorder of interest, and subsequently [b], for the disorders combined. As reviewed above, the high comorbidity in the three disorders and its highly patient-specific and variable pathological development has been only recently acknowledged. As an example, potential categories of tinnitus could comprise of the following characteristics: with predominant mental health issues, with predominant dysfunctional health behaviours, with chronic pain, with hearing loss, with cognitive impairment, or as an isolated condition. It is also highly possible to make a distinction between patients with an overall high vs. low burden of disorder.

As an example, and encouraged by recent recommendations of the IMMPACT association (Edwards et al., 2016), patient-self assessment is favoured whenever available as the basis for phenotyping. It reflects patients’ individual health perceptions and may therefore enlighten mechanisms determining inter-individual differences in health outcomes. If different instruments have been used to assess the same construct, standard item-response theory methods estimating scores on one common scale can be applied [53,77].

An initial approach for phenotyping could be Latent Class Analyses (LCA), as they are highly flexible approaches to include multiple types of variables [78]. LCA are a subset of structural equation modelling, used to find groups or subtypes of cases in different types of data (ordinal, nominal, count, continuous). Observed, multivariate variables are related to a set of latent discrete variables (classes) [79], or as in our case, the assumed phenotypes.

In addition to the unsupervised latent class definition of LCA, which will shed light on specific combinations of variables, it could be utilised to perform a supervised analysis of the data in order to build predictive models for the phenotypes of interest. This approach will be based on classification (e.g., predicting severity levels of the disorder), regression (e.g., predicting the degree of discomfort), or a combination of both. For this, a recent model called TabNet [80] recommends itself. It is based on deep neural networks (DNN) and is specifically adapted to tabular data, which implements an attention mechanism allowing it to reconstruct the most relevant features in a classification or regression task. The attention mechanism implements a “decision-tree-like” mapping of the input features to the labels without the need of performing feature engineering [80]. Hence, unlike traditional “black-box” type DNN-based models, TabNet allows for interpretability of the classification/regression task, in terms of most informative features. Given the flexibility of this model to perform classification and regression tasks, both tasks should be combined into a unified, multi-task learning model. This joint representation learning has been shown to improve performance compared to each task trained separately (single-task learning) for the prediction of depression levels from patient data [81].

Finally, phenotyping based on large numbers of different diagnostic parameters could be done by means of Multidimensional Cluster Statistics (MCS) [82]. There, each diagnostic parameter is treated as an independent dimension, so that the combination of n diagnostic parameters from a given patient can be represented as a point in n-dimensional space. The Euclidian distance between points in that space then is a measure of dissimilarity between patients. Clusters of points formed by several patients’ data can be statistically tested against each other to see if they form specific phenotypes.

In a first effort for phenotyping, a recent study used an analogous data-driven approach to identify multiple patterns of hearing loss in tinnitus patients [83,84]. Analyses using LCA on chronic pain patients revealed four classes separated by the severity of pain and affective symptoms [85]. While this previous work suffers from the limitations imposed by small datasets and a limited focus on the comorbid conditions, it delivers a data-driven proof of concept that a phenotyping approach is feasible.

In addition to the evaluation of the interactive patterns, it will be interesting to determine psychosocial factors across the identified phenotype classes. Several studies related improvement and mitigation of clinical symptoms to the development or strengthening of psychological resilience. An important aspect that seems to contribute to stress vulnerability is the experience of adverse childhood experiences, which can increase the likelihood of developing several mental and physical disorders [86,87,88,89,90]. Whilst psychological risk factors increase an individual’s risk to develop chronic impairment following exposure to biopsychosocial stressors in the absence of psychosocial resources or resilience, the exact interplay of these factors and their respective weightings remain unclear.

In addition to diagnostic measurements, other parameters like lifestyle, social or environmental factors can be included in phenotyping. For example, there is already some evidence that nutrition and nutrients have an impact on the development of tinnitus. As very recent examples, lowered consumption of vitamins B2 and B3, water, and protein correlate with tinnitus and tinnitus distress [91]. Similarly, higher intake of vitamin B12 correlates with lower probability of tinnitus, while higher consumption of calcium, iron, and fat are associated with an increased risk. Moreover, higher vitamin D intake and a diet high in fruit, vegetables, and meat, and low in fat, are associated with a lower risk of hearing difficulties [92]. The briefly reviewed literature here emphasises the need for a combined dataset addressing several significant research questions in a more comprehensive way in the future. These include socioeconomic variables, lifestyle variables, (such as nutrition or leisure time activities), educational status, and other psychosocial variables next to known biological factors to estimate the risks for an onset of a particular collection of symptoms.

## 6. Conclusions: System Medicine Approaches—A Change from Classical Phenotypes to New Phenotypes Based on All Complex Components of Comorbid Disorders

Systems Biology is an approach no longer focussing on individual cellular or molecular components, but on properties of whole systems [93]. Biological organisms should be taken as “a network of interconnected and mutually dependent components that constitute a unified whole” [93]. These biological networks should be analysed at various levels of organization from molecules, cells and organs to organisms and groups of interacting organisms [94]. Systems Medicine takes a Systems Biology perspective by integrating data from various relevant levels of assessment by using modern informational sciences. Consequently, there is a shift from medically reacting to a disorder, to stressing prediction, prevention, personalization and participation. Underlying this approach is the change from classical isolated phenotypes to new phenotypes driven by the modelling of factors of disorders (Figure 3). As proposed for other non-communicable diseases [5], both hypothesis-driven and discovery-driven approaches should complement each other to enable the understanding of the mechanisms, prognosis, and diagnosis, allowing for personalised treatment of disorders. A Systems Medicine approach, as we propose here, is not driven by a priori classification systems, but aims towards the identification of new biomarkers of co-morbidities, disorder severity, and progression. Classical medical markers include blood, saliva, and urine. In a current publication regarding tinnitus, 58 biomarkers were identified concerning endocrine, homeostatic, immunologic, inflammatory, metabolic, neurologic, oxidative, and other parameters [95]. For depression, a meta-analysis demonstrated that peripheral levels of interleukin-6 (IL-6), tumour necrosis factor (TNF)-alpha, IL-10, the soluble IL-2 receptor, C-C chemokine ligand 2, IL-13, IL-18, IL-12, the IL-1 receptor antagonist, and the soluble TNF receptor 2 were increased in patients suffering from major depression compared to controls. In contrast, interferon-gamma levels were reduced in depression (for similar results see [96]).

In addition to such biomarkers, psychological markers are promising to understand complex phenotypes such as chronic pain. Using latent class analyses, Obbarius et al. [85] demonstrated that depression, anxiety, and physical health were markers for the burden of pain. What is currently missing is an overarching approach combining biomarkers and psychosocial markers spanning comorbid disorders. Severity of disease, physical activity, nutritional status and intake, emotional status and wellbeing, quality of life, and lifestyle factors should be included to comprehend the risk factors, the origin, and the pathway of a disorder. The sampling of such biopsychosocial markers will constitute an important step in such an enterprise; meanwhile, the ongoing phenotyping of complex comorbid disorders is amendable as well. Using the example of chronic tinnitus, Niemann et al. [84] identified four subgroups of chronic tinnitus analysing a rather large database using cluster analysis and up-to-date visualization tools. This approach identified four groups namely an “avoidance”, a “psychosomatic”, a “somatic”, and a “distress” group. Regarding chronic pain, Obbarius et al. [85] used a similar approach and identified four phenotypes labelled as “high pain burden”, “extreme pain burden”, “moderate pain burden”, and “low pain burden”. Future research should map the different phenotypical groups with underlying biopsychological profiles of markers. Most helpful for such an approach will be eHealth data addressing physical activity, nutrition, and wellbeing. Obviously, the large amount of data assessed will require modern technological approaches, including the use of AI for analysis. The number of disorders and amount of data necessary can be reached only by including several clinics and centres, ideally spanning several countries. This approach stresses the requirement for high data quality and data security measures as exemplified in the European Health and Evidence Network (https://www.ehden.eu/; accessed on 10 November 2022). This network also proposes open science and open data policies. To achieve this, standards of how data acquisition has to be conducted must be defined and strictly adhered to across clinics and centres involved. For example, for EEG recordings researchers have to agree upon the number of recording channels, recording sites to be included, sampling rate, filter settings, format in which data are saved, data access permissions, and so on. Data analysis, on the other hand, should then be conducted very individually, based on the specific expertise of the respective clinic or centre, to generate added value.

Going beyond the state-of-the-art, future projects should investigate mechanisms involved in the development of chronic comorbid disorders using big data and AI technology for a more profound and precise understanding of the transients of development. Decision support systems based on AI will help combine disorder-specific guidelines, so that they also apply to individual patients suffering from high comorbidity. Identification of such pathways will consequently provide an opportunity for the identification of predictors (e.g., environmental factors, behavioural aspects, biomarkers) and the development of disorder classification. This creates a window of opportunity for prevention and intervention, aiming towards occurrence and overall course of the disorder. Moreover, analysis of pre-existing data with novel data acquired will provide a hitherto unavailable overview of the role and function of specific risk factors involved in the development or chronic manifestation of comorbid disorders. Building an empirically derived treatment decision support system embodied in the medical application has the innovation potential to treat chronic disorders within a personalised medicine treatment approach. Specifically, providing strategies for people at high risk for any comorbid disorders that are currently undetectable by clinicians in clinical practice would support clinicians obtaining a more comprehensive profile of risk factors. Crucially, there is a clear methodological and clinical potential for subsequent future extrapolation to other chronic comorbid disorders, thereby expanding the future scope of research. Additionally, upcoming eHealth applications can empower patients to actively avoid certain behavioural risk factors with the aim of lowering the chance of chronic manifestation of a disease, and can help doctors with their decisions in the near future. Essentially, the innovative nature of a personalised medicine approach to comorbidities is able to lay the fundamental groundwork for clinical implementation of its intrinsically translatable findings, their empirical evaluation, refinement, and dissemination in the 21st century.

## Figures and Tables

**Figure 1 nutrients-14-04320-f001:**
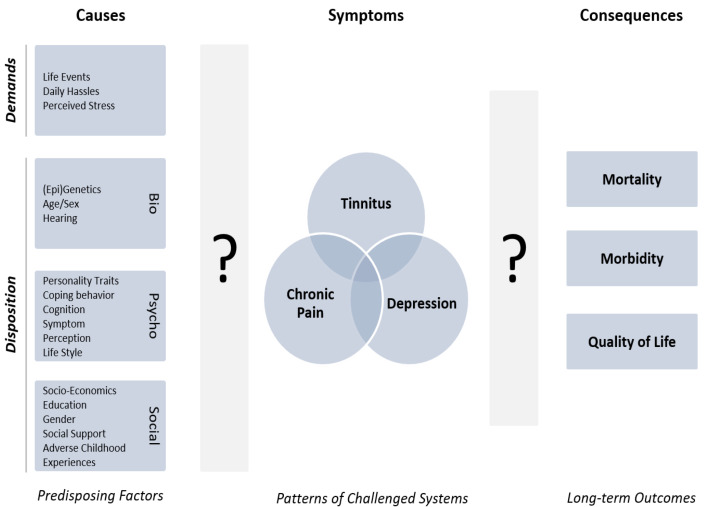
Development of comorbid conditions. Predisposing factors (**left**) interact and result in idiosyncratic maladaptations that manifest in combinations of comorbid conditions. Instead of focussing on classic disorder categories, our approach will identify phenotype classes based on (interactions of) shared and specific risk factors, and use this information to predict symptom co-occurrence and illness trajectories. Question marks indicate that individual pathways from predisposing factors to challenged systems and long-term outcomes are currently not known.

**Figure 2 nutrients-14-04320-f002:**
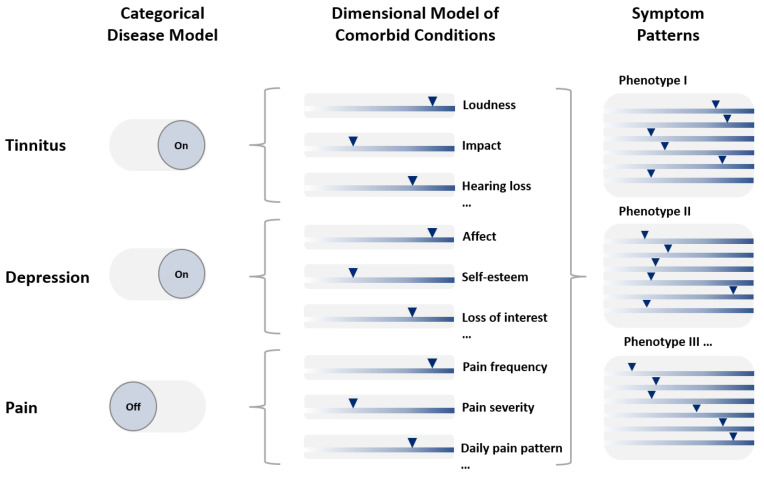
From categorical disorder models to phenotypes of symptom patterns. The same compilation of individual symptoms and their parameter values (**middle** column) would lead to binary categories in the categorical disorder model (**left**), whereas the systems medicine approach favoured here would lead to different phenotypes specific for an individual patient. Note that whereas in the categorical disorder model (in the example given) pain-related symptoms would be neglected, and all data would contribute to the individual phenotypes in the new approach. Triangles depict individual scores for a given dimension or symptom.

**Figure 3 nutrients-14-04320-f003:**
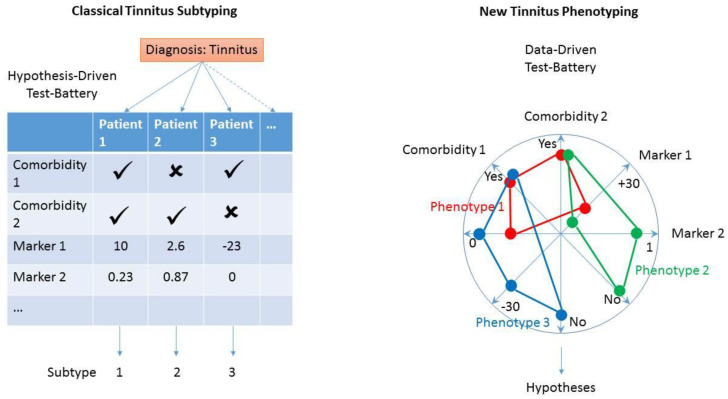
From classical, hypothesis-driven phenotypes to new, data-driven phenotypes. In classical tinnitus subtyping (**left**), tinnitus subtypes are the result of hypotheses assuming tinnitus to be associated with certain comorbidities, facultatively complemented by a number of test markers (presence indicated by ✓, absence indicated by 🗶). In our alternative approach (**right**), tinnitus subtypes result from clustering of an arbitrary number of test markers and diagnostic findings.

**Table 1 nutrients-14-04320-t001:** Prevalence and financial burden of tinnitus, depression, and chronic pain as a proportion of the EU population with the approximate direct and indirect cost imposed on EU societies.

Disorder	Prevalence	Preventable Burden (B EUR/y)
**Tinnitus**	11−30%	117–325
**Depression**	25%	170
**Chronic Pain**	19−30%	400

## Data Availability

Not applicable.

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
