# Peer review of "Systems Medicine Approach for Tinnitus with Comorbid Disorders"

_nutrients, 2022, doi:10.3390/nu14204320_

Round 1

Reviewer 1 Report

This is an innovative approach to tinnitus, pain, and depression, which are all highly correlated. There is only one paragraph addressing nutrients, so I'm not sure it is appropriate for this particular journal, but it should be published in an important journal.

Author Response

Dear Reviewer #1,

We thank you very much for your effort in reading our ms and your highly encouraging response. We wrote a Cover letter addressing the editors and reviewers where we provide a point-by-point rebuttal. You find this letter attached.

We added new paragraph :

. Classical medical markers include blood, saliva and urine. In a current publication re-garding tinnitus, 58 biomarkers were identified concerning endocrine, homeostatic, im-munologic, inflammatory, metabolic, neurologic, oxidative and other parameters. For de-pression, a meta-analysis demonstrated that peripheral levels of interleukin-6 (IL-6), tu-mor necrosis factor (TNF)-alpha, IL-10, the soluble IL-2 receptor, C-C chemokine ligand 2, IL-13, IL-18, IL-12, the IL-1 receptor antagonist, and the soluble TNF receptor 2 were in-creased in patients with suffering from major depression compared to controls. In con-trast, interferon-gamma levels were reduced in depression  (for similar results see also  Köhler, 2017; Vogelzangs, 2016).
In addition to such biomarkers, psychological markers are promising to understand complex phenotypes such as chronic pain. Using latent class analyses, Obbarius et al. (2020) demonstrated that e.g. depression, anxiety and physical health were markers for burden of pain. What is currently missing is an overarching approach combining bi-omarkers and psychosocial markers spanning comorbid disorders. Severity of disease, physical activity, nutritional status and intake, emotional status and wellbeing, quality of life and lifestyle factors should be included to comprehend the risk factors, the origin and pathway of a disorder. The sampling of such biopsychosocial markers will constitute an important step in such an enterprise. On the other hand, phenotyping of complex comor-bid disorders is amendable as well. As an example for chronic tinnitus, Niemann et al. (2020) identified four subgroups of chronic tinnitus analysing a rather large database us-ing cluster analysis and up-to-date visualization tools. This approach identified four groups namely an “avoidance”, a “psychosomatic”, a “somatic” and a “distress” group. Regarding chronic pain, Obbarius et al. (2020) used a similar approach and identified four phenotypes labeled as “high pain burden”, “extreme pain burden”, “moderate pain bur-den” and “low pain burden”. Future  research should map the different phenotyp-ical groups with underlying biopsychological profiles of markers. Most helpful for such an approach will be eHealth data addressing physical activity, nutrition and wellbeing. Obviously, the large amount of data assessed will require modern technological ap-proaches including AI for analysis. The number of disorders and amount of data neces-sary can be reached only by including several clinics and centers, ideally spanning several countries. This approach stresses the requirement for high data quality and data security measures as exemplified in the European Health and Evidence Network (https://www.ehden.eu/). This network also proposes open science and open data policies. To achieve this, standards of how data acquisition has to be conducted must be defined and strictly adhered to across clinics and centers involved. For example, for EEG record-ings researchers have to agree upon the number of recording channels, recording sites to be included, sampling rate, filter settings, format in which data are saved, data access permissions, and so on. Data analysis on the other hand should then be conducted very individually, based on the specific expertise of the respective clinic or center, to generate added value.

Reviewer 2 Report

The authors addressed the systems medicine approach. It is very interesting to define tinnitus, chronic pain, or depression. However, I have curious about how to define markers or phenotypes of diseases in this concept. The text's content is explained in an abstract manner, making it difficult to comprehend. It is better to provide examples of markers or phenotypes of tinnitus or comorbid conditions.     

It is essential to use eHealth, big data, or AI technology to identify predictors for tinnitus and its comorbidities. However, it is critical to select appropriate quality data for analysis. The authors have to describe the methodology of systems medicine approach it in a more specific way. It is unclear how the examination will proceed based solely on the current description of the systems medicine approach for tinnitus.  

Author Response

Comment:

"The authors addressed the systems medicine approach. It is very interesting to define tinnitus, chronic pain, or depression. However, I have curious about how to define markers or phenotypes of diseases in this concept. The text's content is explained in an abstract manner, making it difficult to comprehend. It is better to provide examples of markers or phenotypes of tinnitus or comorbid conditions.”

Response:
We included a longer paragraph in the Conclusions section with new literature where we specify bio- and psychosocial markers for tinnitus, depression and pain. We agree with the reviewer that this will make the ms more readable.

Comment:

“It is essential to use eHealth, big data, or AI technology to identify predictors for tinnitus and its comorbidities. However, it is critical to select appropriate quality data for analysis. The authors have to describe the methodology of systems medicine approach it in a more specific way. It is unclear how the examination will proceed based solely on the current description of the systems medicine approach for tinnitus."

Response:

We fully agree with the reviewer and we stress the importance of eHealth, big data, or AI technology in more detail now. In particular we explain exemplarily for EEG data how standards have to be defined for data acquisition for all groups within a consortium, but that analysis that uses such data can (and should) be very individual based on the respective expertise of each group to generate added value. As above, this can be found now in the Conclusions section.

We want to thank the reviewers for their valuable comments and suggestions. They improved our manuscript significantly.

Round 2

Reviewer 2 Report

The authors have answered the issues pointed out.

However, it does not seem to properly present the phenotype for tinnitus or depression.

Furthermore, it is unclear how the authors' depression-related biomarkers can be used to diagnose depression, because biochemistry tests are not performed to diagnose depression in general.

Many unnecessary tests are required for system medicine, but I'm not sure if this is a problem.

It appears that it is necessary to consider whether many unnecessary tests are required to classify phenotype of those diseases.

Author Response

We thank the reviewer for her/his encouraging evaluation and the emphasis that we addressed all open issues.

The comment of the reviewer gives us the impression that we have a different definition for phenotype. In our understanding, phenotype addresses all variables indexing observable characteristics and traits of an organism and thus, biomarkers, neurophysiological measures, but also lifestyle factors such as nutrition habits belong to the phenotype.

In this vein, biomarkers should be regarded as additional measures for diagnostic procedures allowing the detection of potential subtypes. In the sense of a personalized medicine approach, the inclusion of biomarkers is currently mandatory. Along these lines, it is not possible to classify a given test for e.g. a biomarker as necessary or unnecessary before extensive research was conducted.

We hope that we responded satisfactorily to the comment of the reviewer and that she/he can follow our argument. Currently, we would therefore not do more changes in our manuscript.